# Prescriber-Initiated Engagement of Pharmacists for Information and Intervention in Programs of All-Inclusive Care for the Elderly

**DOI:** 10.3390/pharmacy8010024

**Published:** 2020-02-21

**Authors:** David L. Bankes, Richard O. Schamp, Calvin H. Knowlton, Kevin T. Bain

**Affiliations:** Applied Precision Pharmacotherapy Institute, Tabula Rasa HealthCare, Inc., Moorestown, NJ 08057, USA; RSchamp@CPSTN.com (R.O.S.); cknowlton@trhc.com (C.H.K.); KBain0225@gmail.com (K.T.B.)

**Keywords:** drug information, collaboration, pharmacist, Programs of All-Inclusive Care for the Elderly, medication safety

## Abstract

Little is known about the types of drug information inquiries (DIIs) prescribers caring for older adults ask pharmacists during routine practice. The objective of this research was to analyze the types of DIIs prescribing clinicians of Programs of All-Inclusive Care for the Elderly (PACE) made to clinical pharmacists during routine patient care. This was a retrospective analysis of documented pharmacists’ encounters with PACE prescribers between March through December, 2018. DIIs were classified using a developed taxonomy that describes prescribers’ motivations for consulting with pharmacists and their drug information needs. Prescribers made 414 DIIs during the study period. Medication safety concerns motivated the majority of prescribers’ inquiries (223, 53.9%). Inquiries received frequently involved modifying drug therapy (94, 22.7%), identifying or resolving adverse drug events (75, 18.1%), selecting or adjusting doses (61, 14.7%), selecting new drug therapies (57, 13.8%), and identifying or resolving drug interactions (52, 12.6%). Central nervous system medications (e.g., antidepressants and opioids), were involved in 38.6% (n = 160) of all DIIs. When answering DIIs, pharmacists made 389 recommendations. Start alternative medications (18.0%), start new medications (16.7%), and change doses (12.1%) were the most frequent recommendations rendered. Prescribers implemented at least 79.3% (n = 268) of recommendations based on pharmacy records (n = 338 verifiable recommendations). During clinical practice, PACE prescribers commonly ask pharmacists a variety of DIIs, largely related to medication safety concerns. In response to these DIIs, pharmacists provide medication management recommendations, which are largely implemented by prescribers.

## 1. Introduction

It is estimated that prescribers caring for older adults (PCOAs) raise about two clinical questions for every patient seen during an encounter, and drug information typically accounts for the largest proportion of these questions [1]. Despite considering the majority of their questions to be “important” for patient care, PCOAs have indicated that they do not find or pursue answers to about 50% of their questions [1]. Writing prescriptions in spite of unmet drug information needs can lead to poor care quality and adverse drug events, both of which disproportionately affect older adults [1,2,3,4,5,6]. As drug information experts, pharmacists can help solve this potential problem. Broadly, evidence indicates that when prescribers have an opportunity to collaborate with pharmacists, they pose drug information inquiries (DIIs) [7,8,9,10,11]. Moreover, information-seeking behavior can improve prescribing practices and patient outcomes, such as mortality [12,13,14,15]. 

Presently, little is known about the characteristics of inquiries that PCOAs pose to pharmacists during outpatient practice. A recent study of DIIs posed by PCOAs to pharmacists at a Norwegian drug information center found that about 80% of inquiries were equally divided between adverse drug effects and drug treatment considerations [9]. However, this study involved complex inquiries that do not necessarily represent the types of questions that arise during routine practice, and this study’s findings may not adequately represent the drug information needs of PCOAs practicing in the United States [9]. Other recent investigations into this subject matter exist but did not analyze DIIs made specifically during the care of older adults [7,8,16,17,18]. 

Obtaining a better understanding of the drug information needs of PCOAs would shed light on the supportive role of pharmacists as allied healthcare professionals and may identify opportunities for pharmacists to collaborate with PCOAs in geriatric care. Since inquiries not only represent genuine needs but also suggest knowledge gaps, there may be implications regarding continuing education for PCOAs and/or pharmacists. The objective of our study was to analyze the types of DIIs that prescribers made to pharmacists during the routine care of community-dwelling older adults enrolled in the Programs of All-Inclusive Care for the Elderly (PACE). Our secondary objectives were to compare the drug information needs of non-physician prescribers (e.g., nurse practitioners) to physicians, to appraise recommendations that pharmacists made in response to those DIIs, and to determine the implementation of these recommendations by PACE prescribers.

## 2. Materials and Methods 

### 2.1. Study Design 

This was a retrospective analysis of pharmacy records from a national provider of PACE pharmacy services. This study was granted a waiver of informed consent by the Biomedical Research Alliance of New York Institutional Review Board (protocol number 19-12-072-420, approved 27 February 2019). Researchers conducted the study in accordance with the ethical principles set forth by the Declaration of Helsinki.

### 2.2. Setting and Context 

PACE is a government-funded service that is an alternative to long-term care institutionalization [19]. To enroll in PACE, participants must be 55 years or older, live in the PACE service area, and be certified by their state to need a “nursing home level of care”, yet be able to live safely in their community through PACE [19]. Therefore, PACE participants are a cohort of community-dwelling older adults with complex medical and medication needs. According to the National PACE Association, the average PACE participant is 76-years old and has six chronic medical conditions (vascular disease, diabetes with chronic complications, dementia, depression, bipolar, polyneuropathy, and congestive heart failure are among the top chronic conditions of PACE participants) [20]. Collaboration with an interdisciplinary team (IDT) comprised of various healthcare professionals is a cornerstone of PACE [19]. The IDT coordinates care planning and determines the frequency at which participants travel to PACE centers for their healthcare needs [19]. The number of prescribers at PACE varies by the organization’s census. For example, an organization with a smaller census (e.g., <120 participants) may have one physician and one nurse practitioner, whereas a larger organization (e.g., >220 participants) may have several physicians, nurse practitioners, and/or physician assistants. For the latter, in particular, a team-based approach to individual participant care is often utilized. PACE centers often partner with a single pharmacy provider for all of their medication services (e.g., dispensing, consulting) [21]. At a minimum, the pharmacy must dispense both prescription and over-the-counter medications for PACE participants [19]. Currently, regulations do not require pharmacists to be part of the IDT [19], but many PACE organizations contractually collaborate with one or more pharmacists as part of their care model [21,22,23].

The pharmacy involved in this study is a centralized pharmacy operating from three locations in the United States. It provides comprehensive medication services for more than 35 PACE organizations, including about 75 individual PACE centers, representing a net census exceeding 10,000 PACE participants. In addition to dispensing medications, the pharmacy provides clinical consulting services to help prescribers optimize medication regimens and reduce medication-related risks for PACE participants [22,24]. The pharmacy employs board-certified geriatric pharmacists (i.e., BCGP) who use a clinical decision support system to identify and mitigate medication-related problems, including simultaneous multi-drug interactions [22]. The clinical pharmacists collaborate with PACE organizations, namely prescribers, to manage the day-to-day medication needs for their participants. As part of this collaboration, prescribers make DIIs to pharmacists. The DIIs are usually submitted via telephonic and electronic (i.e., encrypted e-mail and instant message) communication methods because pharmacists are not routinely on site at the PACE centers.

### 2.3. Data Source and Study Sample

This study used a convenience sample that included PACE participants with documented DIIs within the pharmacy records between 1 March 2018 and 31 December 2018. DIIs were obtained from two data sources in the pharmacy records: clinical encounter logs and secure instant message archives. Duplicate instant message DIIs that were documented in both data sources were excluded. To ensure that DIIs were organic and originated from PACE prescribers, we excluded inquiries that prescribers made as follow-up to either a pharmacist- or prescriber-initiated intervention, inquiries that originated from PACE staff without prescriptive authority (e.g., nurse or medical assistant) and technical- or system-related questions pertaining to product identification, product selection, or order entry. Inquiries whose details were insufficiently documented to allow for adequate categorization were also excluded.

### 2.4. Procedures and Definitions 

We developed a taxonomy to classify DIIs into mutually exclusive categories, rooted in the medication-related problem that could occur if the inquiry remained unanswered. DIIs were characterized by both the prescriber’s primary motivation for collaboration with the pharmacist and the prescriber’s specific information need. This taxonomy is defined in Table 1. DII encounters were culled from the pharmacy records by using various search terms and filters highly likely to be associated with DIIs at this practice site (e.g., advise, asked, interaction, question, recommend, request, review, suggest, question; see Appendix A for complete search strategy). Next, the details of each encounter were evaluated, and DIIs were classified, according to the aforementioned taxonomy, by the primary researcher. If the primary researcher was unable to make a classification decision, the question was classified by the senior researcher. DIIs were tagged by type of prescriber (i.e., physician or non-physician prescriber); method of DII communication (i.e., telephonic or electronic, whereby electronic included e-mails and instant messages); and drug class or classes referenced by the prescriber in the DII. For each prescriber, we estimated the years of practice experience by years enumerated with a National Provider Identifier (NPI) issued by Centers for Medicare and Medicaid Services. For DIIs that yielded specific and actionable pharmacist recommendations, the associated recommendations were classified according to a modified version of a well-recognized taxonomy [25]. To gauge the impact of prescriber–pharmacist collaboration on patient care, pharmacy records were reviewed for evidence of recommendation implementation within 90 days of the DII response. We allowed a protracted follow-up to account for DIIs that may have involved complex disease state management or multi-phase care plans. Recommendations that could not be verified for implementation via pharmacy records (e.g., laboratory monitoring) were excluded from the implementation analysis. 

### 2.5. Statistical Considerations

Descriptive statistics were used to report DIIs, drug classes, recommendations made by pharmacists to prescribers, and recommendations implemented by prescribers. The chi-square test was used to analyze two-group comparisons for nominal data. Specifically, comparisons were made in 2 × 4 and 2 × 8 contingency tables to assess for differences between physicians and non-physician prescribers in motivation for collaboration and specific information needs, respectively. An alpha level of 0.05 was set to determine statistical significance. Analyses were performed using Microsoft Excel (Microsoft 2013, Redmond, WA, USA).

## 3. Results

During the 10 month study period, 584 encounters were identified as possible prescriber-initiated DIIs. After exclusion criteria were applied, 414 encounters were determined to be DIIs that originated organically from a PACE prescriber and were included in the analytical sample. Of these inquiries, 312 (75.4%) originated electronically and 102 (24.6%) originated telephonically. Figure 1 depicts data management for the ascertainment of DIIs.

Table 2 describes the demographic characteristics of PACE participants, prescribers, and sites. In sum, 23 clinical pharmacists responded to DIIs made by 102 PACE prescribers from 59 PACE sites of varying census sizes. Overall, 65.7% (n = 67) of PACE prescribers were non-physician prescribers. The overwhelming majority of non-physician prescribers were nurse practitioners (n = 65, 97.0%) and most had 10 or fewer years of practice experience (n = 53, 79.1%). PACE physicians had more medical practice experience, with 77.1% (n = 27) practicing for 11 or more years. During the study time frame, each prescriber made, on average, 4.1 (±5.4) inquiries (median = 2, interquartile range [IQR] = 1,5). PACE prescribers issued unique DIIs for 359 PACE participants. On average, each patient had 1.2 ± 0.4 DIIs (range 1–4 DIIs) raised by prescribers regarding their care.

Table 3 reports PACE prescribers’ primary motivations for asking DIIs, the information needs requested, and the drug classes referenced by prescribers during the inquiry. Overall, 53.9% (n = 223) of DIIs were motivated by medication safety concerns. Medication effectiveness concerns were also common (n = 107, 25.8%). Adherence (n = 54, 13.0%) and cost (n = 30, 7.2%) were less common motivators. There was no statistically significant difference between physicians and non-physician prescribers in their motivations for asking DIIs (P = 0.41). Across all DIIs, prescribers referenced 53 drug classes. Drugs affecting the central nervous system, particularly antidepressants, opioids, anticonvulsants, antipsychotics, anxiolytics, and sedative hypnotics, were frequently referenced (n = 137, 33.1% for the aforementioned classes). Additionally, drugs affecting the cardiovascular system, particularly antihypertensives, hyperlipidemia agents, antiplatelets and anticoagulants, as well as antidiabetic agents, were also commonly referenced by prescribers making inquiries (n = 89, 21.5% for the aforementioned classes).

The collection of PACE prescribers’ information needs was marked by heterogeneity, as no single information need clearly predominated or exceeded 25%. The top five most frequent information needs, which represented 81.9% of all inquiries, were for help with modifying existing drug therapy (n = 94, 22.7%), identifying or managing adverse events and side effects (n = 75, 18.1%), making dose selections or dose adjustments (n = 61, 14.7%), formulating new drug therapy selections (n = 57, 13.8%), and identifying or managing drug interactions (n = 52, 12.6%). While 64.0% (n = 265) of DIIs were asked by non-physician prescribers, there was no statistically significant difference in their information needs as compared with the needs of physicians (P = 0.09). However, there was a tendency for non-physician prescribers to ask more questions related to therapy modifications, adverse effects, and doses compared to their physician counterparts.

In response to prescribers’ DIIs, pharmacists rendered 389 recommendations. The number of recommendations was less than the total number of DIIs because some inquiries involved no intervening recommendation (e.g., a prescriber asked if any profiled medications can cause a precipitous decline in glomerular filtration rate) and others did not have an associated recommendation documented by the pharmacist (e.g., prescriber asked for an opioid tapering schedule but the specifics of the taper were not documented in pharmacy records). As demonstrated in Table 4, the recommendations varied based on the specific inquiry and included starting alternate drugs (n=70, 18.0%), starting new drugs (n = 65, 16.7%), altering an existing dose (n = 47, 12.1%), starting or adjusting a drug at a specific dose (n = 42, 10.8%), and discontinuing medications (n = 33, 8.5%). Of the recommendations that could be verified for implementation using pharmacy records (n = 338), prescribers implemented 79.3% (n = 268). Physicians and non-physician prescribers implemented recommendations at similar frequencies (physicians, 78.6% versus non-physician prescribers, 79.7%).

## 4. Discussion

Over the 10-month study period, geriatric clinical pharmacists responded to more than 400 DIIs from approximately 100 PACE prescribers throughout the United States. The DIIs primarily pertained to safety and effectiveness concerns and frequently involved drugs from the central nervous system, cardiovascular, and endocrinology classes. While specific question types were wide-ranging, prescribers often inquired about selecting and modifying therapy, and identifying and managing adverse effects and drug interactions. The characteristics of DIIs were found to be similar between non-physician prescribers and physicians. As a result of these encounters, pharmacists commonly provided recommendations to change or alter drug regimens, which were usually—about 80%—implemented by PACE prescribers. Collectively, these results indicate that, through collaboration with prescribers, pharmacists can directly influence drug therapy decisions for PACE participants.

The overwhelming majority of DIIs were motivated by prescribers’ concerns related to safety and effectiveness and involved clinical decisions for drug treatment (e.g., altering doses, selecting medications) and patient management (e.g., managing adverse effects). Therefore, when PCOAs were faced with drug-related uncertainties, pharmacists were frequently recruited to help make medication management decisions and to prevent medication-related problems. This is not surprising. When interviewed in qualitative studies, physicians and other healthcare practitioners have acknowledged that although medication management for older adults is complex and challenging, it can be optimized through interdisciplinary collaboration with pharmacists [26,27,28]. Moreover, other studies of this subject matter similarly suggested that the majority of drug information needs of PCOAs involve adverse drug reactions and drug selection considerations [1,9]. Our study’s findings shed much-needed light on the drug information needs of PCOAs and, moreover, the collaboration that transpires between prescribers and pharmacists in the PACE setting.

The DIIs that pharmacists in this practice setting addressed are often associated with avoidable healthcare costs. As an example, an economic evaluation of pharmacy services conducted in a Veterans Affairs setting suggested that 400 to 2000 USD in additional outpatient costs can be avoided for each resolved untreated or undertreated indication, adverse reaction, inappropriate dose, or drug interaction [29]. The drug classes most commonly encountered by pharmacists responding to DIIs in this study are frequently associated with drug-related emergency department visits and hospitalizations [4,5,30,31,32]. Older adults are particularly susceptible to the adverse effects of opioids, anticoagulants, and antidiabetic agents, and these drugs have been implicated in increased utilization of healthcare resources, such as emergency department visits, amongst this population [4,5]. While more research is needed to determine whether pharmacist-provided drug information to PCOA results in improved patient outcomes, we showed that pharmacists’ recommendations were implemented at a high frequency (80%) and resulted in drug regimen alterations. If pharmacists are providing accurate answers, these findings suggest that the medication-related problems that might arise from prescribing with information deficits can be avoided. 

We also found that inquiries made by non-physician prescribers and physicians were not significantly different in terms of motivations and needs, which suggests that the value of pharmacist collaboration may transcend both prescriber role and practice experience. In our study sample, nearly 80% of PACE physicians had at least a decade of practice experience, suggesting that even experienced physicians have legitimate needs for collaboration with pharmacists. Conversely, nearly half of all non-physician prescribers in our study sample had less than five years of practice experience. Compared with physicians, non-physician prescribers receive less training on drug pharmacology and have historically reported relying on pharmacists to provide drug information and assist with prescribing [10,33,34]. Roles and experience aside, most PACE prescribers strictly care for older adults, yet they still actively engaged pharmacists when faced with drug-related uncertainties. Therefore, prescribers caring for all ages (e.g., family practice physicians), who could be less familiar with the intricacies of geriatric pharmacotherapy, may have even more questions than their PACE counterparts. If so, this study would underestimate the value of collaboration with pharmacists specializing in geriatrics and could suggest a greater need for collaboration beyond geriatric-predominant settings. 

This investigation confirms that drug utilization in geriatrics is full of uncertainties and that pharmacists can play a vital role in helping to manage medications for PCOAs. Specific to PACE, our findings could be the impetus for regulators to revise the PACE model in order to make pharmacists requisite members of the IDT [19]. More generally, the rapidly aging population [35] is creating demands for geriatricians that are far outpacing the stagnating supply of new specialists [36]. As the burden of care shifts to prescribers who might be less well-versed in geriatric pharmacotherapy principles, increased collaboration with pharmacists will be needed. No matter the care setting, pharmacists must be prepared for new collaborative endeavors in geriatric care. In the practice setting of this study, 100% of pharmacists are required to be board certified in geriatrics and, furthermore, are supported by a state-of-the-art clinical decision support system [22,24,37,38]. Such extensive training in geriatric pharmacotherapy, coupled with technology support, likely helped these pharmacists respond to DIIs from prescribers. The demonstrated competency of these pharmacists inevitably built trust with prescribers, which is critical to ensuring successful collaborative relationships [39,40,41]. Nevertheless, economic, clinical, and humanistic outcomes associated with such collaborations were not assessed in this study and these outcomes require future investigation. 

This study has important limitations that may have underrepresented DIIs or biased our data. First, it is possible that the search of pharmacy records was insufficient to capture all DIIs during the 10 month study window. Second, the centralized nature of the pharmacy excludes face-to-face collaboration. It is possible that our remote collaboration method influenced the total number or characteristics of DIIs received. Encounters initiated over the telephone or through personal e-mail must be manually documented by a pharmacist during workflow, which may discourage pharmacists from logging DIIs that they deem clinically insignificant. About 50% of the total sample of DIIs were culled solely from instant message archives and yet were not documented in the pharmacist encounter log, suggesting under-documentation. Nevertheless, this study’s sample of questions was roughly four times larger than the prior studies assessing general or drug-specific information needs of PCOA [1,9]. While larger samples are desirable, we believe that our sample size provides meaningful results given the number of pharmacists (23) and prescribers (102) involved in this study. Third, manually documented encounters are recorded through the lens of the documenting pharmacist. This may have led to misclassified prescriber motivations or information needs and—while unlikely—falsely identified DIIs are a possibility. However, 71% of all questions were secure instant messages, which permitted researchers the opportunity to read these DIIs verbatim. Manually documented encounters with ambiguous details were flagged then discussed with the documenting pharmacist. If pertinent details could not be obtained from the pharmacist, the encounter was excluded from the dataset. Finally, regarding prescriber experience, NPI enumeration dates may imperfectly estimate the years of prescriber practice experience. 

## 5. Conclusions

During clinical practice, PACE prescribers commonly ask pharmacists a variety of DIIs, largely related to medication safety. In response to these DIIs, pharmacists provide medication management recommendations, which are often implemented by prescribers. Further research is needed to fully evaluate the economic, clinical, and humanistic outcomes associated with the provision of drug information in PACE.

## Figures and Tables

**Figure 1 pharmacy-08-00024-f001:**
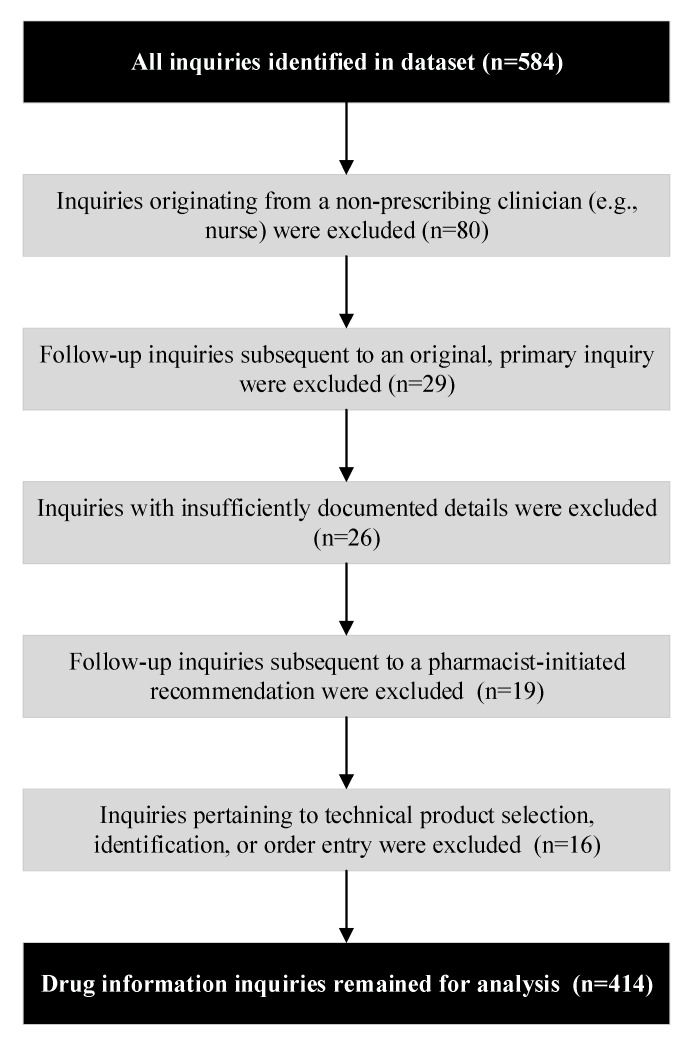
Data Management and Workflow Diagram for Ascertainment of Drug Information Inquiries.

**Table 1 pharmacy-08-00024-t001:** Taxonomy Used to Characterize Drug Information Inquiries.

**Motivation:**	**A Prescriber’s Primary Motivation for Contacting a Pharmacist is:**
effectiveness	for help treating a medical condition that either (a) has not yet undergone treatment or (b) has not been adequately treated.
safety	to avoid, manage, mitigate, or identify actual or potential drug toxicity or unintended adverse reactions that may be in the context of clinical or laboratory abnormalities, drug allergies, unnecessary drug therapy, or drug interactions.
adherence	to resolve known or suspected (a) non-adherence, (b) non-receipt of a medication (e.g., inability to swallow, patient refusal of certain medications), or (c) risk factors that may cause non-adherence (e.g., improving a needlessly complex drug regimen).
cost	for assistance with pharmacotherapeutic decisions based on price quotes, pharmacoeconomic data, or formulary restrictions.
**Information Need:**	**A Prescriber Needs a Pharmacist to:**
modification of existing drug therapy	advise how to add to or change the existing therapy because the underlying disease has not been optimally treated, has been over-treated, has not been cost-effectively treated, or has been potentially unsafely treated with the current drug regimen.
dose selection or adjustment	advice regarding adjusting, selecting, tapering, or cross-tapering doses.
adverse events and side effects	identify, manage, or explain actual or suspected side effects or clinical abnormalities related to suspected drug toxicity; provide advice to change therapy to resolve the actual or potential adverse event/side effect.
new drug therapy selection	choose a drug for a new, untreated indication.
drug interactions	identify, manage, or explain drug-drug, simultaneous multi-drug, drug-disease, drug-gene, or drug-food interactions.
price quote	provide pricing information to help make a treatment decision.
monitoring parameters	advise how to monitor drug therapy for safety and/or efficacy or to interpret or explain aberrations in laboratory/clinical data, where aberrations are not suspected to be related to an adverse event.
general drug information	provide general drug information (pharmacology, contraindications/warnings, mechanism of action, storage conditions, active ingredients, intravenous flow rates, timing of administration, ability to crush/split, product availability, etc.) for educational purposes in order to inform decision making.

**Table 2 pharmacy-08-00024-t002:** Study Demographics.

Characteristic	Value *
**Participants**	**359**
Age, mean ± SD (range)	72.9 ± 9.8 (55–98)
Female	251 (69.9)
Medications, mean ± SD (range)	16.6 ± 6.6 (1–42)
Ethnicity	
Caucasian	77 (21.4)
African American	37 (10.3)
Hispanic	5 (1.4)
Asian	2 (0.6)
American Indian	2 (0.6)
Unspecified	236 (65.7)
DIIs per patient, mean ± SD (range)	1.2 ± 0.4 (1–4) ^†^
**Prescribers**	**102**
DIIs per prescriber, mean ± SD (range)	4.1 *±* 5.4 (1–29)
DIIs per prescriber, median (IQR)	2 (1.5)
**Physicians**	**35 (34.3)**
Female	21 (60.0)
≤5 years in practice	1 (2.9)
6–10 years in practice	7 (20.0)
≥11 years in practice	27 (77.1)
**Non-physician Prescribers**	**67 (65.7)**
Female	64 (95.5)
Nurse Practitioner	65 (97.0)
Physician Assistant	2 (3.0)
≤5 years in practice	32 (47.8)
6–10 years in practice	21 (31.3)
≥11 years in practice	14 (20.9)
**Pharmacists**	**23 ^‡^**
Female	16 (69.6) ^‡^
DIIs per pharmacist, mean ± SD (range)	18.0 *±* 18.1 (1-65) ^‡^
DIIs per pharmacist, median (IQR)	12 (4,25) ^‡^
**PACE Sites**	**59**
US Region	
Northeast	28 (46.7)
South	14 (23.3)
Midwest	12 (20.0)
West	5 (8.3)
**Census**	
<120 participants	21 (35.6)
120–220 participants	25 (42.4)
>220 participants	13 (22.0)

* Values represented as n (%), unless otherwise stated. ^†^ 48 participants had more than one question. Of these, seven had two different prescribers each ask at least one unique question regarding the participant’s care. The remaining 41 had the same prescriber ask more than one unique question regarding the participant’s care. ^‡^ One DII had missing pharmacist data and data associated with this DII were excluded in measures of central tendency and overall pharmacist count. Abbreviations: PACE = Programs of All-Inclusive Care for the Elderly; DIIs = drug information inquiries.

**Table 3 pharmacy-08-00024-t003:** Drug Information Inquiries from Programs of All-Inclusive Care for the Elderly (PACE) Prescribers to Clinical Pharmacists.

	All Prescribers	Physicians	NPPs	P-Value *
Drug Information Inquiries, n (%)	414 (100)	149 (36.0)	265 (64.0)	
**Primary Motivation for Inquiry**				0.41
Safety	223 (53.9)	84 (56.4)	139 (52.5)	
Effectiveness	107 (25.8)	39 (26.2)	68 (25.7)	
Adherence	54 (13.0)	14 (9.4)	40 (15.1)	
Cost	30 (7.2)	12 (8.1)	18 (6.8)	
**Information Need**				0.09
Modifications of existing drug therapy	94 (22.7)	30 (20.1)	64 (24.2)	
Adverse events and side effects	75 (18.1)	23 (15.4)	52 (19.6)	
Dose selections or adjustments	61 (14.7)	16 (10.7)	45 (17.0)	
New drug therapy selections	57 (13.8)	24 (16.1)	33 (12.5)	
Drug interactions	52 (12.6)	28 (18.8)	24 (9.1)	
General drug information	48 (11.6)	18 (12.1)	30 (11.3)	
Price quotes	17 (4.1)	6 (4.0)	11 (4.2)	
Monitoring parameters	10 (2.4)	4 (2.7)	6 (2.3)	
**Drug Class Referenced by Prescriber ^†^**				
Antidepressants	49 (11.8)	20 (13.4)	29 (10.9)	
Antidiabetic agents	35 (8.5)	8 (5.4)	27 (10.2)	
Opioid analgesics	30 (7.2)	15 (10.1)	15 (5.7)	
Antibiotics	28 (6.8)	14 (9.4)	14 (5.3)	
Antihypertensives	28 (6.8)	6 (4.0)	22 (8.3)	
Anticonvulsants	26 (6.3)	8 (5.4)	18 (6.8)	
Antipsychotics	15 (3.6)	5 (3.4)	10 (3.8)	
Hyperlipidemia agents	14 (3.4)	6 (4.0)	8 (3.0)	
Inhaled agents for COPD/asthma	13 (3.1)	2 (1.3)	11 (4.2)	
Antiplatelets/anticoagulants	12 (2.9)	4 (2.7)	8 (3.0)	
Anxiolytics	11 (2.7)	7 (4.7)	3 (1.1)	
Antifungals	10 (2.4)	5 (3.4)	5 (1.9)	
Antisecretory agents	9 (2.2)	2 (1.3)	7 (2.6)	
Vitamins/minerals	9 (2.2)	4 (2.7)	5 (1.9)	
Natural products	8 (1.9)	2 (1.3)	6 (2.3)	
Antigout agents	7 (1.7)	2 (1.3)	5 (1.9)	
PDE inhibitors	6 (1.4)	2 (1.3)	4 (1.5)	
Sedative hypnotics	6 (1.4)	2 (1.3)	4 (1.5)	
Urinary incontinence agents	5 (1.2)	2 (1.3)	3 (1.1)	

Abbreviations: PACE = Programs of All-Inclusive Care for the Elderly; NPPs = non-physician prescribers (i.e., nurse practitioner or physician assistant); COPD = chronic obstructive pulmonary disease; PDE = phosphodiesterase. * Comparison between physicians and NPPs in the categorical distributions for motivation and need, using Chi-square tests as 2 × 4 and 2 × 8 contingency tables, respectively. ^†^ A total of 53 drug classes were referenced by PACE prescribers. Those referenced in ≤1.0% of DIIs were not tabulated.

**Table 4 pharmacy-08-00024-t004:** Recommendations Made by Pharmacists in Response to Drug Information Inquiries from PACE Prescribers.

Recommendation	n (%)
Start alternative therapy	70 (18.0)
Start or restart medication	65 (16.7)
Dose change (reduce or increase)	47 (12.1)
Start/alter therapy at specific dose	42 (10.8)
Discontinue medication	33 (8.5)
Laboratory or symptom monitoring	33 (8.5)
Dosage form change	29 (7.5)
Specific taper/titration plan	20 (5.1)
Confirm a prescriber’s plan	18 (4.6)
Schedule change	15 (3.9)
No change in therapy	10 (2.6)
Duration of treatment change	4 (1.0)
Hold medication	2 (0.5)
Specialist referral	1 (0.3)
**Total**	389 (100)

Abbreviations: PACE = Programs of All-Inclusive Care for the Elderly.

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
