# Peer review of "Prescriber-Initiated Engagement of Pharmacists for Information and Intervention in Programs of All-Inclusive Care for the Elderly"

_pharmacy, 2020, doi:10.3390/pharmacy8010024_

Round 1

Reviewer 1 Report

Overall, this manuscript answers a key question:  "What do PACE providers need from a pharmacist?", which is a question that many potential PACE pharmacists struggle with answering.  The paper is well-written.  Here are some specific comments:

Section 2.2:  Were there no face-to-face DIIs? Based on the paragraph, it seems that it was only telephone or email/IM.  The next section seems to imply that these DIIs were captured as part of the EMR or other records, so possibly face-to-face DIIs were just not captured?

Table 2 contains data from the patients involved in the study.  However, the information in the introduction discusses demographics for all of PACE.  It was not immediately clear that these were different, and I was initially confused as to why the data were different.  Suggest clarifying either in the intro or on the table.

For table 4, it seems to me that the *description below the chart should be included as part of the narrative for that section as opposed to a footnote for the table.

In the discussion, the sentence that begins with "Fortunately, our study showed that recommendations..." seems to imply the that medication-related ADEs can be prevented through what happened in the study.  While I agree with this statement in general, the data here do not include outcomes related to hospitalizations, mortality, disease state control, etc.  Therefore, I suggest rewording this sentence to be less leading toward the assumption that a pharmacist intervention = better outcomes.  

In the discussion section, the sentence that begins with "One may surmise that..." is controversial.  I think one could easily argue that the opposite is true (experience = more knowledge).  I am not familiar with the citation at the end of the sentence, but even if that particular study shows this is possible, it is still not a fair statement to present it this way (one-sided).  

In the discussion section, the sentence that discusses making a pharmacist a requisite member of the IDT -  YES!!!  I say again, YES!!

One question that I am left wondering after reading this is why so many questions arise in an encounter with an older adult (as described in the introduction), but so few were identified as DII in the study?  I think that this should be a larger part of the discussion.  We are moving in the right direction, but how much more could we be doing as pharmacists?  I have a similar practice site (provider of 30+ years, one clinical pharmacist, site 130 ppts) and I document 150+ DIIs each year, all of which are face-to-face requests.  If all 100+ sites had this type of utilization, the data would be enormous.  I'm rambling, but this could be a powerful discussion point for moving forward.

Reviewer 2 Report

Thank you for a very interesting paper, I found it interesting and more over  very important for practice. Below some question which arose when I was reading the paper:

do the patients in PACE system have one or more professionals, who can prescribe medication for them, maybe authors could add some information about it in the 2.2 section. When I analysed the data in the table 2, it was shown that 102 prescribers made DIIs for 356 patients, but it did not explain if each DII was made for different patient or if it was possible that some of DIIs were made for the same patients by the same/or different prescriber, maybe author could give a short comment on it below the table 2 I did not found any details about clinical pharmacists, who offered the service for prescribers, how many pharmacists was involved, maybe some data about them could be added in table 2
